# WAVEDIF: Wavelet sub-band based Deepfake Identification in Frequency Domain

## Abstract

001 *With the more realistic convergence of Deepfakes, its' iden-*
002 *tification becomes more demanding. Recently, numerous*
003 *deepfake detection techniques have been proposed, most of*
004 *which are in the spatio-temporal domain. While these meth-*
005 *ods have shown promise, many of them neglect convinc-*
006 *ing artifacts that exhibit different patterns across frequency*
007 *domains. This research proposes WAVEDIF, a strict fre-*
008 *quency domain, lightweight deepfake video detection al-*
009 *gorithm using wavelet sub-band energies. In WAVEDIF,*
010 *for feature extraction, each video undergoes a Discrete*
011 *Fourier Transform to filter out high-frequency noisy de-*
012 *tails (quite evident in deepfakes). These representations are*
013 *then decomposed into their respective wavelet sub-bands –*
014 *LL (Low-Low), LH (Low-High), HL (High-Low), and HH*
015 *(High-High) passing them through a Haar Filter, following*
016 *which the energy values (particular to each sub-band) are*
017 *computed. These energy values are then used to learn a lin-*
018 *ear decision boundary (using regression analysis), which is*
019 *then used for classification. This enables an interpretable,*
020 *lightweight deterministic technique for the detection of syn-*
021 *thesized videos, besides achieving an accuracy compara-*
022 *ble to the state-of-the-art. Experimental results on popu-*
023 *lar deepfake video datasets shows over 92% accuracy for*
024 *in-dataset evaluation, and 88% accuracy for cross dataset*
025 *evaluation.*

## 1. Introduction

027 Deepfakes are artificially generated videos in which the fa-
028 cial expression or contours of any source is replaced or
029 transformed or concatenated with that of a target subject
030 [6, 18, 20, 24, 34]. Deepfakes, in recent times are becom-
031 ing very realistic, and thus can have severe negative societal
032 impact [2, 9], and thus necessitates the need for having de-
033 tection mechanism to identify such manipulated media.

034 In recent years, deepfake research has taken into con-
035 sideration two different domains of operation (primarily)

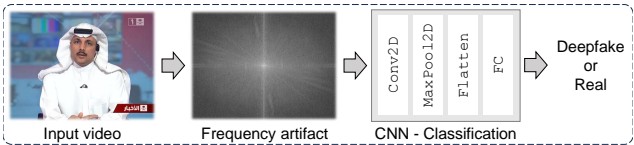

(a) Conventional frequency-based Deepfake Identification

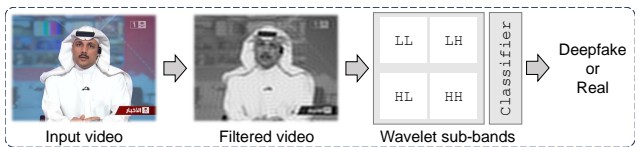

(b) WAVEDIF: Wavelet energy-based Deepfake Identification

Figure 1. WAVEDIF: Wavelet sub-band based Deepfake Identification in Frequency Domain. (a) Traditionally, frequency level artifacts (like DFT, DCT, FFT, etc.) are generated, which in image format are fed to Convolutional Neural Networks (CNNs) for feature extraction, and then classification. (b) WAVEDIF filters out high frequency artifacts using a low-pass gaussian filter and uses DWT to decompose videos into sub-bands LL, LH, HL, and HH. Further, the energy level values of these sub-bands are used as input features for classification.

036 – the **spatial** domain [1, 4, 11, 19], which involves pixel-
037 based manipulations, facial landmarks, and texture synthe-
038 sis, and the **frequency** domain [10, 12, 15, 28], which in-
039 volves signal transformations, frequency artifacts, and in-
040 consistencies in high and low-frequency details. Addition-
041 ally, some works [3, 31, 33] have suggested multi-modal
042 deepfake detection, where features from more than one do-
043 main are combined and fused to form yet another complex
044 feature set, based on which the video is classified as original
045 or deepfake.

046 Traditionally, features from all (or a set of) domains are
047 feed to deep learning modules which learns domain-specific
048 artifacts, therefore enabling a classification. While many
049 works from the recent literature considers spatial domain
050 for deepfake detection, comparatively fewer contributions
051 have been made towards deepfake detection strictly per-
052 taining to the frequency domain. This research presents

| Original | DEEPFAKE | FACE2FACE | FACESHIFTER | FACESWAP | NEURALTEXTURES |
|---|---|---|---|---|---|

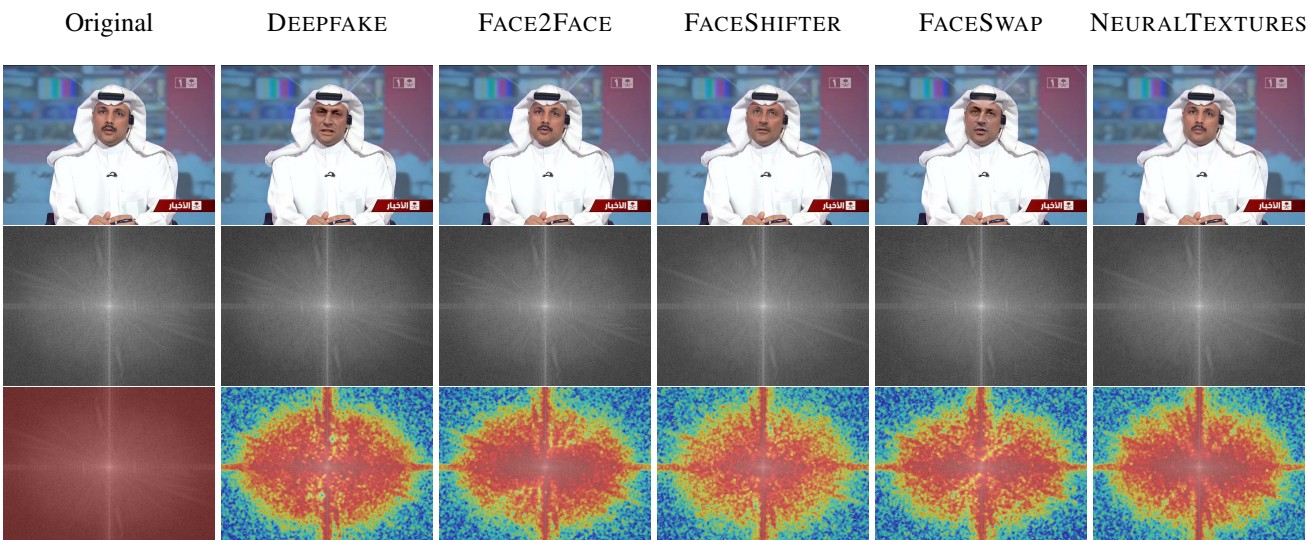

Figure 2. A sample video from the `FaceForensics++` [25] dataset (first row). Corresponding to the selected sample, five deepfakes – DEEPFAKES [13], FACE2FACE [29], FACESHIFTER [16], FACESWAP [21], and NEURAL TEXTURES [30] are generated. The second row shows the Discrete Fourier transformation of the frames. Though difficult to visualize, the DFTs differ from each other considerably; SSIM (Original, DEEPFAKES) = 0.6028, SSIM (Original, FACE2FACE) = 0.6010, SSIM (Original, FACESHIFTER) = 0.5811, SSIM (Original, FACESWAP) = 0.5823, SSIM (Original, NEURAL TEXTURES) = 0.5991. The last row shows the difference heatmap where bright red or yellow areas denotes regions of major modifications, in green or blue are regions of moderate to low modifications.

WAVEDIF, a strict frequency domain, lightweight deepfake video detection algorithm using wavelet sub-band energies (LL, LH, HL, and HH are the sub bands). Every deepfake detection framework usually works in two phases – (a) *feature extraction*, wherein features particular to original and deepfake videos are learned, and (b) *classification*, wherein based on the learned feature a decision boundary is laid between deepfake and original videos [6]. Conventional deepfake detection frameworks mostly rely on convolutional neural networks (CNNs) for the feature extraction process, and a fully-connected layer is maintained at the end for the classification phase. While deep learning-based feature extraction and classification are often very accurate, most of them usually require strong computational power for their perusal. In contrast to these traditional deep learning-based frameworks, in WAVEDIF, the feature extraction phase is based on wavelet sub-band energies extracted through Discrete Wavelet Transform (DWT), which enables decomposition of the video frames into different frequency components while preserving spatial locality. This further allows us to simultaneously examine both high and low-frequency artifacts introduced during deepfake synthesis. Prior to wavelet decomposition, Discrete Fourier Transform (DFT) for each frame of the input video is performed, which filters out high frequency artifacts that get added to the videos during deepfake synthesis.

Fig. 2 shows the result of applying DFT for selected frames of videos from the `FaceForensics++` dataset [25]. It also shows the difference heatmap (corresponding to original and different deepfake representations) to elucidate the relevance of of our approach of classifying deepfakes in the frequency domain. The classification phase in WAVEDIF relies on the computed (sub-band - LL, LH, HL, and HH) energy values, based on which a decision boundary (along with a threshold) is learned through regression analysis. Classification of new (unseen) video examples is through the application of these decision boundary. Fig. 3 shows the DWT sub-bands' three-dimensional visualization (for a selected video pair from `FaceForensics++` ,i.e., original and corresponding five deepfakes, where the $x$, and $y$ axes represents the spatial dimensions, and $z$ axis represents the wavelet coefficients' magnitude. This enables pictorial visualization of the features used for classifying the videos, and the marked differences (with yellow dots) justifies the relevance of wavelet sub-bands' energies for the classification.

A common practice with frequency domain deepfake detection models is to utilize high-frequency artifacts for the classification. **In contrast, in this research high-frequency details are filtered out, since wavelet decomposition is a multi-resolution analysis which is very sensitive to noise and high-frequency distortions**. In particular, to get rid of spurious high-frequency noise, Gaussian low-pass filter was used. Direct utilization of the DWT coefficients for classification leads to poor localization in the frequency domain due to the widespread noise across all

sub-bands.

To evaluate the effectiveness of the proposed WAVEDIF technique, two popular deepfake (video-only) datasets – `FaceForensics++` [25], and `CelebDF (v2)` [17] have been considered. The evaluation results shows competitive performance by WAVEDIF compared to state-of-the-art deep learning based deepfake detection frameworks across all domain, **while incurring much lesser computational cost. WAVEDIF achieves $\approx 94.93\%$ in-dataset, and $\approx 88.83\%$ cross-dataset accuracies for the `FaceForensics++` dataset. Similarly, for the `CelebDF (v2)` dataset, the metrics are** $\approx 92.03\%$**, and** $\approx 87.01\%$ **respectively.**

To sum up, the novelties of the proposed WAVEDIF methodology in contrast to existing frameworks are as follows:

1. Existing deepfake detection techniques mostly rely on features extracted from the spatial (standalone) or fusion features from spatial domain with those from other domains like audio, spectra, etc. [3, 19, 31], but WAVEDIF operates strictly on features extracted from the frequency domain.

2. Existing techniques mostly rely on deep learning-based feature extraction, making them computationally expensive (hidden layers like attention [8, 14] and convolution [22, 26] are computationally expensive). In contrast, WAVEDIF uses Discrete Wavelet Transformation (DWT) to decompose video frames into LL, LH, HL, HH sub-bands, and then to their respective energies $\mathcal{E}_{LL}$, $\mathcal{E}_{LH}$, $\mathcal{E}_{HL}$, and $\mathcal{E}_{HH}$, which makes the feature extraction stage lightweight.

3. Existing techniques mostly rely on fully-connected layers for classification [5, 23], which have hereby been replaced by linear and logistic regression to model an interpretable decision boundary with a threshold. This makes the classification both lightweight and interpretable.

The rest of the manuscript is organized as follows – Section 2 presents related works on deepfake detection (across the two primary domains – spatial, and frequency). Section 3 explains in detail the proposed methodology - WAVEDIF. Section 4 is directed towards evaluating the proposed methodology, experimenting with deepfake datasets, comparing the performance of WAVEDIF with respect to the state-of-the-art detection frameworks. Finally, the paper concludes with comments about direction of future work in Section 5.

## 2. Related Works

The domain of deepfake detection has evolved into a multifaceted area of research. This section discusses research on deepfake forensics from the recent past across spatial and frequency domain. Spatial domain methods analyze pixel-level inconsistencies, frequency domain methods analyze artifacts in spectral features (like DCT/DFT/FFT, etc.) While spatial domain (sometimes also fused with other domains like auditory domain [3, 31, 33]) are dominant methods for detection, frequency-based approaches are gaining prominence due to their lightweight feature extraction process, and invariance to adversarial shifts.

### 2.1. Spatial Domain Approaches

Spatial methods primarily rely on Convolutional Neural Networks (CNNs) and Transformer-based architectures to capture features specific to deepfakes, and using those features to classify between real and synthetic videos (mainly done using fully-connected layers). The features include pixel-level manipulations, facial landmark distortions, and texture inconsistencies introduced in the synthetic videos generated using Generative Adversarial Networks (GANs). Naskar *et al.* [19] proposed a spatial domain deepfake detection approach using *deep feature stacking* and *meta-learning* integrating features extracted by XCEPTION and EFFICIENTNET-B7 through a stacking-based ensemble framework. The extracted features are further selected using a multi-layer perceptron meta-learner for classification. Agarwal *et al.* [1] proposed a multi-domain *cross-stitched network* for deepfake detection –MD-CSDNETWORK. It combined spatial and frequency domain features to improve generalization. The model has two parallel branches – for processing spatial information and frequency-domain artifacts present in fake videos. Das *et al.* [4] proposed a masked autoencoding spatiotemporal transformer-based deepfake detection method using *self-supervised learning*. The model combined two VISION TRANSFORMERS – *Spatial Transformer* that learns frame-level visual features from individual RGB frames, and *Temporal Transformer* for learning motion inconsistencies (by analysing optical flow fields). He *et al.* [11] proposed GAZEFORENSICS, that uses gaze-guided spatial inconsistency learning (e.g. unnatural eye movements) for improving deepfake detection accuracy. They used *3D gaze estimation network* to extract gaze representations, which are then used for classification by integrating consistency between real and fake gaze patterns.

### 2.2. Frequency Domain Approaches

Frequency domain analysis based methods analyze the spectral properties of the videos, capturing frequency artifacts that generative models unintentionally introduce (due to inconsistencies in texture synthesis). The features generally include Discrete Fourier Transformation (DFT), Discrete Cosine Transformation (DCT), Fast Fourier Transform (FFT), etc. Tan *et al.* [28] proposed FREQNET, a frequency-aware deepfake detection framework designed for better generalization across different deepfake generation models. While traditional methods detect artifacts in-

| Input Video | LL (*Low-Low*) | LH (*Low-High*) | HL (*High-Low*) | HH (*High-High*) |
|---|---|---|---|---|

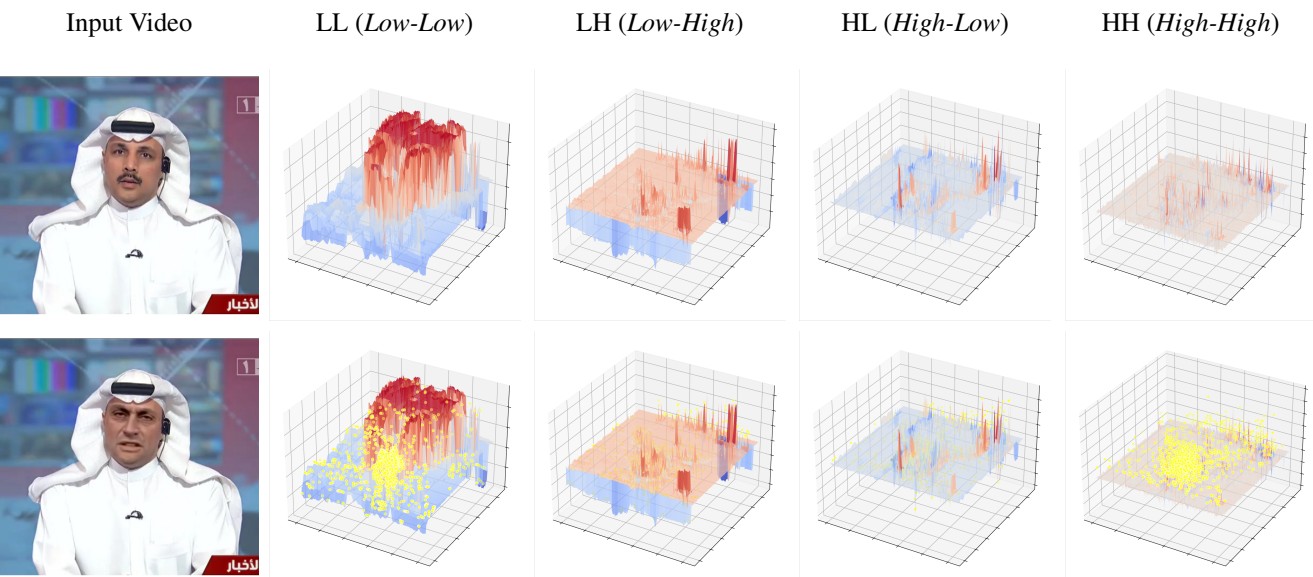

Figure 3. Motivation for using wavelet sub-bands (and their energies) as a distinguishable feature between real and deepfake video. The **first row** corresponds to energy sub-band visualization (in 3D) for a **real video**, while the **second row** corresponds to visualization (in 3D) for a **synthetically prepared video**. Note that in the visualizations of the second row, yellow dots represent coefficients with difference in magnitude.

troduced during the up-sampling process in GAN pipelines, FREQNET uses frequency domain learning by applying convolutional layers to the phase and amplitude spectra between Fast Fourier Transform (FFT) and Inverse FFT (iFFT). Kohli and Gupta [15] proposed a frequency-based convolutional neural network (fCNN) for detecting DEEP-FAKE, FACESWAP, and FACE2FACE facial forgeries (particular to as seen in `FaceForensics++` dataset). They convert the facial images from each of these classes to their respective frequency domain using two-dimensional *Global Discrete Cosine Transforms* (2D-GDCT), which are then processed using a three-layer *Frequency CNN* (fCNN) to learn and therefore classify between real and fake faces. Hasanaath *et al.* [10] introduced Frequency-Enhanced Self-Blended Images (FSBI) that integrates *self-blended images*, and *frequency-domain analysis*. For conversion to frequency domain, Discrete Wavelet Transform (DWT) was used on the self-blended images, which are then used to extract features from, by using convolutional neural network (standard for frequency-based feature extraction). Jeong *et al.* [12] proposed FREPGAN using *frequency-level perturbation maps*. The training process in FREPGAN is divided into two phases – *Early Training* (which identifies frequency-level artifacts), and *Later Training* (which identifies higher-level inconsistencies).

## 3. Proposed Methodology: WAVEDIF

This section presents WAVEDIF (Wavelet sub-band based Deepfake Identification in Frequency Domain), the proposed methodology for deepfake detection in this research. Since WAVEDIF takes the wavelet sub-band energies as feature input to the model, the model operates strictly in the frequency domain. WAVEDIF captures the subtle frequency artifacts that are introduced in videos (especially in their spectral domain) during artificial synthesis, and classifies the videos accordingly.

Given an input video (say $v \in \mathbb{R}^{H \times W \times 3 \times F}$), the objective is to classify whether the video is synthetically generated or original; where, $H \times W$ is the spatial resolution of the video, the factor three denotes three-channel RGB representation, and $F$ corresponds to the frame count of the video. The proposed WAVEDIF model is a two-stage-pipeline – (i) *feature extraction*, which captures the frequency domain abnormalities, and (ii) *classification based on the extracted features*, to train a deterministic model which can classify between real and deepfake videos. Additionally, the feature extraction phase is further subdivided into two parts – (a) *DFT-based frequency filtration*, which enhances the discriminative abnormalities (patterns) by suppressing noise and (b) *DWT-based feature extraction*, which extracts localized frequency variations, based on wavelet sub-band energies. Note that prior to passing any input video to the feature extraction phase, it is converted to grayscale (single-channel), and to maintain uniformity across all videos, they are resized to a fixed preset resolution.

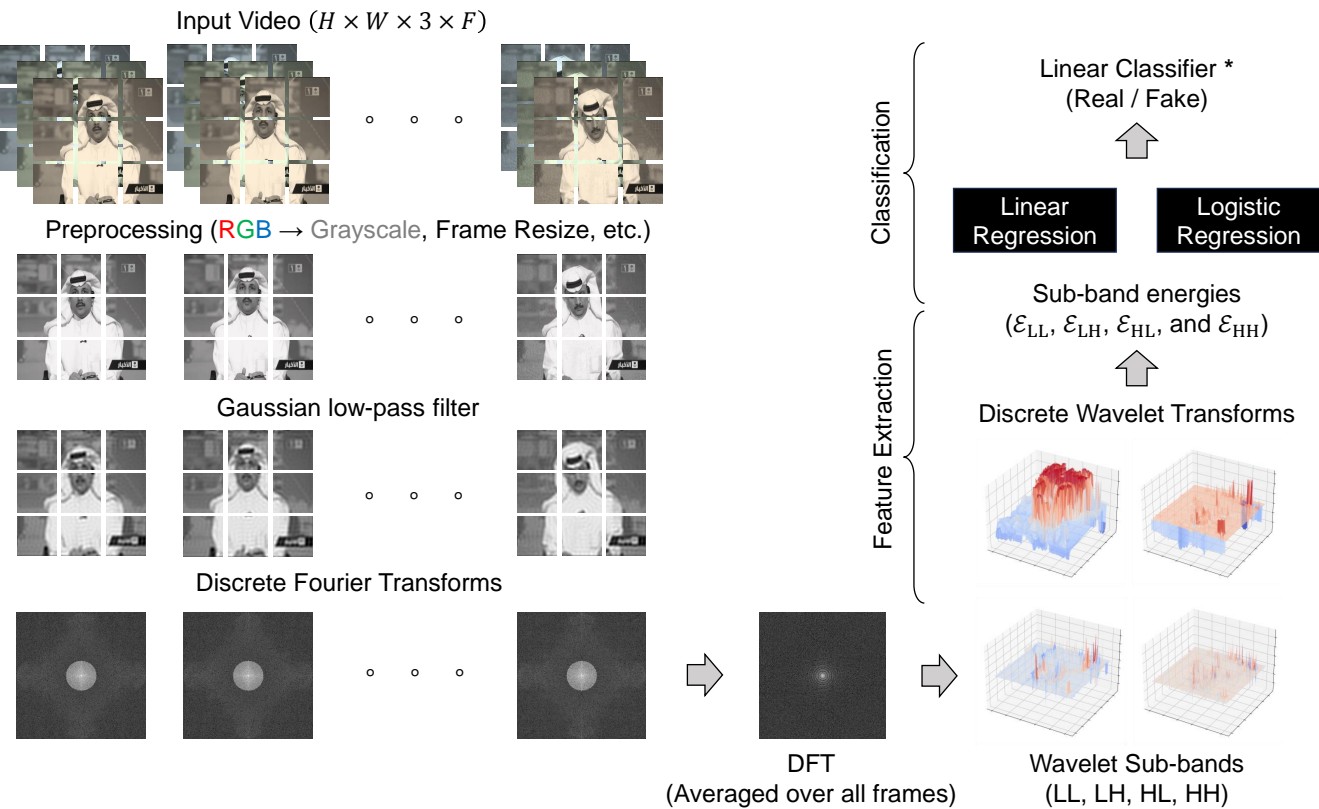

Figure 4. **Architectural Overview of the WAVEDIF Deepfake Detection Technique.** Given a video input $\mathcal{V} \in \mathbb{R}^{H \times W \times 3 \times F}$, each frame of size $(H \times W)$ undergoes a Discrete Fourier Transform (DFT) to filter out high-frequency noise artifacts. The DFTs of all frames are averaged to generate a final DFT representation of the input video $\mathcal{V}$. This representation is then decomposed into wavelet sub-bands (LL, LH, HL, HH) using a Haar filter. Further, the energy values $\mathcal{E}_{LL}$, $\mathcal{E}_{LH}$, $\mathcal{E}_{HL}$, and $\mathcal{E}_{HH}$ are computed corresponding to each video. At the end of the *feature extraction* process (iff phase == TRAINING), a linear decision boundary-based equation is modelled (using linear, and logistic regression). Models pertaining to the correct size of a boundary are *classified* therefore.

## 3.1. Feature Extraction

In the first stage of the WAVEDIF pipeline, initially every input video $v$ is decomposed into the constituent frames, $\{f_1, f_2, \ldots, f_F\}$. Following the decomposition, each frame $f_i$ is first transformed into their frequency domain using the 2D Discrete Fourier Transform (DFT), following Eqn. (1):

$$\mathcal{F}_i(u,v) = \sum_{x=0}^{H-1} \sum_{y=0}^{W-1} f_i(x,y) e^{-j2\pi \left( \frac{ux}{H} + \frac{vy}{W} \right)} \tag{1}$$

where $(x, y)$, and $(u, v)$ are the coordinates in spatial and spectral domain respectively. As mentioned previously, deepfakes often introduce high frequency artifacts which are irrelevant for frequency-based classification [7]. To filter out these noisy artifacts, Gaussian Low-Pass Filtering (GLPF) [32] has been used (as per Eqn. (2)) to suppress the high-frequency components, while preserving discriminative patterns in lower frequencies, which was the recon-structed using Inverse DFT (as per Eqn. (3)):

$$\mathcal{F}_i'(u,v) = \mathcal{F}_i(u,v) \cdot e^{-\frac{(u^2+v^2)}{2\sigma^2}} \tag{2}$$

$$f_i'(x,y) = \sum_{u=0}^{H-1} \sum_{v=0}^{W-1} \mathcal{F}_i'(u,v) e^{j2\pi \left( \frac{ux}{H} + \frac{vy}{W} \right)} \tag{3}$$

where $\sigma$ is the cutoff-frequency. In this work, we set $\sigma = 45$.

To obtain a global spectral representation of the video $(v)$, mean aggregated representation across all frames is computed as, $\mathcal{R} = \frac{1}{F} \sum_{i=1}^{F} f_i'$. Since, DFT captures global frequency information, but lacks spatial localization, DWT using Haar wavelet filter was applied on the aggregated frame $\mathcal{R}$ [27], which decomposed $\mathcal{R}$ into four frequency sub-bands:

1. LL (Low-Low), which captures low-frequency structures.

---

**Algorithm 1** WAVEDIF

**Require:** Labeled dataset $\{\mathbb{V}, \ell\}$, and test video $\mathcal{V} \notin \mathbb{V}$
**Ensure:** Predicted label, $l \in \{\text{ORIGINAL}, \text{DEEPFAKE}\}$
1: **function** FEATUREEXTRACTION $(v)$
2:      $\{f_1, f_2, \ldots, f_F\} \leftarrow$ EXTRACTFRAMES $(v)$
3:      **for** each frame $f_i$ **do**
4:          $\mathcal{F}_i \leftarrow$ DISCRETEFOURIERTRANSFORM $(f_i)$
5:          $\mathcal{F}'_i \leftarrow$ GAUSSIANLPF $(\mathcal{F}_i)$
6:          $f'_i \leftarrow$ INVERSEDFT $(\mathcal{F}'_i)$
7:      **end for**
8:      $\mathcal{R} \leftarrow \frac{1}{F} \sum_{i=1}^{F} f'_i$ ▷ Averaged (for all filtered frames)
9:      $\begin{bmatrix} \text{LL} & \text{LH} \\ \text{HL} & \text{HH} \end{bmatrix} \leftarrow$ DISCRETEWAVELETTRANS-FORM$(\mathcal{R})$
10:      Compute $\mathcal{E}_{\mathcal{S}} = \sum_{j \in \mathcal{S}} \mathcal{S}_j^2, \forall \mathcal{S} \in \{\text{LL}, \text{LH}, \text{HL}, \text{HH}\}$
11:      $\mathbf{F}_v \leftarrow [\mathcal{E}_{\text{LL}}, \mathcal{E}_{\text{LH}}, \mathcal{E}_{\text{HL}}, \mathcal{E}_{\text{HH}}]$ ▷ Feature vector (for $v$)
12: **end function**
13: **if** phase == TRAINING **then** ▷ Training Phase
14:      **for** each video $v$ in $\mathbb{V}$ **do**
15:          $\mathbf{F}_v \leftarrow$ FEATUREEXTRACTION $(v)$
16:          $\mathbf{F}_{\mathbb{V}} \leftarrow \mathbf{F}_{\mathbb{V}} \oplus \mathbf{F}_v$ ▷ Feature fusion
17:      **end for**
18:      Learn the model parameters (**Linear Regression**)

     $\mathcal{B}(\mathbf{F}_{\mathbb{V}}) = \theta_1 \cdot \mathcal{E}_{\text{LL}} + \theta_2 \cdot \mathcal{E}_{\text{LH}} + \theta_3 \cdot \mathcal{E}_{\text{HL}} + \theta_4 \cdot \mathcal{E}_{\text{HH}} + \beta$

19:      Learn the threshold $T$ (**Logistic Regression**)
20: **else** ▷ Inference Phase
21:      $\mathbf{F}_{\mathcal{V}} \leftarrow$ FEATUREEXTRACTION $(\mathcal{V})$
22:      $\{\Theta^{\text{T}}, \beta\} \leftarrow \mathcal{B}(\mathbf{F}_{\mathbb{V}})$
23:      $f(\mathbf{F}_{\mathcal{V}}) \leftarrow \Theta^{\text{T}} \mathbf{F}_{\mathcal{V}} + \beta$
24:      **if** $f(\mathbf{F}_{\mathcal{V}}) \geq T$ **then**
25:          $l \leftarrow$ ORIGINAL
26:      **else**
27:          $l \leftarrow$ DEEPFAKE
28:      **end if**
29:      **return** $l$
30: **end if**

---

2. LH (Low-High), which captures horizontal high-frequency details.
3. LH (High-Low), which captures vertical high-frequency details.
4. HH (High-High), which captures diagonal high-frequency details.

For the Haar wavelet transform, low-pass filters $\left(\phi = \frac{1}{\sqrt{2}}[1, 1]\right)$, and high pass filters $\left(\psi = \frac{1}{\sqrt{2}}[1, -1]\right)$ are applied separately along rows and columns of the matrix $\mathcal{R}$. The ' respective sub-bands are computed as per Eqn. (4):

$$LL(m, n) = \frac{1}{4} \sum_{i=0}^{1} \sum_{j=0}^{1} \mathcal{R}(2m+i, 2n+j)$$

$$LH(m, n) = \frac{1}{4} \sum_{i=0}^{1} \sum_{j=0}^{1} (-1)^j \mathcal{R}(2m+i, 2n+j)$$

$$HL(m, n) = \frac{1}{4} \sum_{i=0}^{1} \sum_{j=0}^{1} (-1)^i \mathcal{R}(2m+i, 2n+j)$$

$$HH(m, n) = \frac{1}{4} \sum_{i=0}^{1} \sum_{j=0}^{1} (-1)^{i+j} \mathcal{R}(2m+i, 2n+j)$$

$$(4)$$

where, $(m, n)$ are the coordinates of transformed domain.

Each sub-band captures some specific frequency response, notably the deepfake videos captures some unnatural energy distributions in these sub-bands, which serves as the basis of classification in this research. The energy corresponding to each of these bands were computed following Eqn. (5):

$$\mathcal{E}_{\mathcal{S}} = \sum_m \sum_n \mathcal{S}^2(m, n), \quad \forall \mathcal{S} \in \{\text{LL}, \text{LH}, \text{HL}, \text{HH}\} \quad (5)$$

or in simpler terms, $\mathcal{E}_{\mathcal{S}} = \sum_{j \in \mathcal{S}} \mathcal{S}_j^2, \forall \mathcal{S} \in \{\text{LL}, \text{LH}, \text{HL}, \text{HH}\}$

Thus, the feature vector corresponding to the input video $v$ is $\mathbf{F}_v = [\mathcal{E}_{\text{LL}}, \mathcal{E}_{\text{LH}}, \mathcal{E}_{\text{HL}}, \mathcal{E}_{\text{HH}}]$.

The feature extraction stage of the WAVEDIF pipeline is common to both training and inference phases. In the training phase, the labeled dataset $\{\mathbb{V}, \ell\}$ is fed for the feature extraction, and based on the features, a deterministic linear boundary equation is trained where $\mathbb{V}$ is a vector of videos, and $\ell$ is the vector of labels corresponding to every video in $\mathbb{V}$. In the inference phase, an unseen video $\mathcal{V} \notin \mathbb{V}$ is taken as input, its features extracted, is passed through the learned boundary equation to predict a label $l \in \{\text{ORIGINAL}, \text{DEEPFAKE}\}$.

### 3.2. Classification based on the extracted features

In the second stage of the WAVEDIF pipeline, the learned features are used for training a deterministic model (if phase == TRAINING), and the trained deterministic model is used for giving a verdict for any input video (if phase == INFERENCE). During training, features corresponding to each video $v$ in the labeled dataset $\mathbb{V}$ are learned, and fused together to form the model feature vector as in Eqn. (6):

$$\mathbf{F}_{\mathbb{V}} = \bigoplus_{v \in \mathbb{V}} \mathbf{F}_v, \quad \mathbf{F}_v = [\mathcal{E}_{\text{LL}}, \mathcal{E}_{\text{LH}}, \mathcal{E}_{\text{HL}}, \mathcal{E}_{\text{HH}}] \quad (6)$$

Next, the model feature vector, and the associated labels $(\mathbf{F}_{\mathbb{V}}, \ell)$ are used for training a uni-dimensional regression

model, i.e., using $(\mathbf{F}_\mathbb{V}, \ell)$ to learn the model weights and biases as per Eqn. (7):

$$\mathcal{B}(\mathbf{F}_\mathbb{V}) = \theta_1 \cdot \mathcal{E}_{LL} + \theta_2 \cdot \mathcal{E}_{LH} + \theta_3 \cdot \mathcal{E}_{HL} + \theta_4 \cdot \mathcal{E}_{HH} + \beta \quad (7)$$

Further, using the same set of features and labels, a logistic regression model was trained to obtain a threshold $(T)$. Thus, the trained model (inclusive of threshold) is as per Eqn. (8):

$$l = \begin{cases} \texttt{ORIGINAL}, & \text{if } \Theta^{\mathrm{T}}\mathbf{F}_\mathcal{V} + \beta \geq T \\ \texttt{DEEPFAKE}, & \text{otherwise.} \end{cases} \quad (8)$$

In the inference phase, for an unseen video $\mathcal{V} \notin \mathbb{V}$, the trained model (Eqn. (8)) is used. Fig. 4 gives a pictorial illustration of WAVEDIF, and Algorithm 1 summarizes the workflow.

## 4. Experimental Results

This section discusses the experimental details, and the results obtained by comparing the proposed WAVEDIF model with state-of-the-art deepfake detection techniques.

### 4.1. Dataset

To evaluate the performance of WAVEDIF, it was tested on the `FaceForensics++` [25], and `CelebDF (v2)` [17] dataset. `FaceForensics++` consists of 1000 real and corresponding 5000 synthetic videos (from five different deepfake generational models). The reason behind the choice of `FaceForensics++` is the presence of synthetic videos from multiple techniques such as FACE2FACE [29], FACESWAP [21], NEURAL TEXTURES [30], etc. `CelebDF (v2)` consists of 590 real and 5639 synthetic videos of celebrities in various lighting conditions, angles, and expressions. The reason behind the choice of `CelebDF (v2)` is the specific designing paradigm of the dataset, which reduce visual artifacts (in both spatial, and spectral domain) commonly found in synthetic videos, thus makes the deepfakes look much more realistic and therefore challenging to detect [17]. Additionally, since the objective of this research is to detect deepfakes strictly from artifacts in frequency domain, videos (with no audio) were chosen as is the property of each video in `FaceForensics++`, and `CelebDF (v2)` dataset.

Fig. 5 shows *t-distributed Stochastic Neighbor Embedding* (t-SNE) representation (as suggested by Naskar *et al.* [19]) of the four-dimensional feature vectors $\mathbf{F}_v = [\mathcal{E}_{LL}, \mathcal{E}_{LH}, \mathcal{E}_{HL}, \mathcal{E}_{HH}]$ for original and deepfake videos from `FaceForensics++`, and `CelebDF (v2)`. It can be interpreted from the visualization that `FaceForensics++` have well separated clusters, thus classification will be more accurate than `CelebDF (v2)` where there are few overlapping clusters. Note that, these features are log-transformed (refer to Subsection 4.2 for the reason)

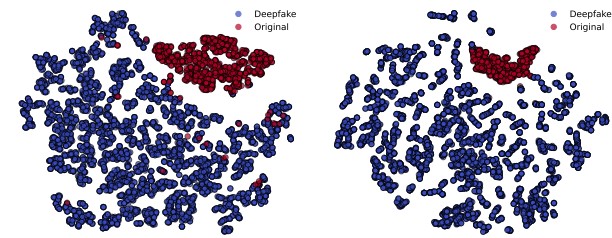

Figure 5. t-SNE visualization (corresponding to wavelet subbands' energies, as features) for both the deepfake datasets - `FaceForensics++` (left sub-figure), and `CelebDF (v2)` (right sub-figure)

.

### 4.2. Experimental Setup

The WAVEDIF model was trained and evaluated on these datasets using an 80-20 train-test split. Prior to extraction of frequency domain feature from the videos, for each frame three channels of input spectra (RGB) were converted to a single channel (Grayscale), and since videos (in real time) can vary in spatial resolution, the videos were resized to $224 \times 224 \times F$ ($F$ is the frame count). Further, since in the classification phase, input to the regression models are sub-band energy values, which are of order $10^8 - 10^9$, feeding them directly to the model might lead to numerical instability (like arithmetic overflow or precision errors). To get rid of that, a log-transformation $\text{log-transform}(z) = \log(1 + z)$ was applied. Note that logarithmic transformation works only for features having values $\geq 0$. Since energy values are computed as sum of squared terms (refer to Eqn. (5)), the features in $\mathbf{F}_v = [\mathcal{E}_{LL}, \mathcal{E}_{LH}, \mathcal{E}_{HL}, \mathcal{E}_{HH}]$ will be $\geq 0$; thus, logarithmic transformation is applicable. All experiments were carried out on a system with 16 GiB main memory, `Intel(R) Core(TM) i7-1065G7 @1.30 GHz` processor and an `NVIDIA GeForce MX330` Graphics Processing Unit with 2 GiB in-built memory.

### 4.3. Evaluation Results

The accuracy of WAVEDIF was compared against a number of state-of-the-art models that work both in the frequency and spatial domains (refer to Section 2). Further, to test the generalizability of the proposed model, it was evaluated both in-dataset and cross-dataset, and their respective accuracies were noted. One advantage of classifying between deepfakes and original videos in the frequency domain is the lightweight model requirements, but because of less complex model artifacts, it trades off the accuracy (though negligibly).

The evaluation results of WAVEDIF on the `FaceForensics++` and `CelebDF (v2)` datasets have been presented in Table 1, and 2 respectively. In

Table 1. Comparison of performance between different models operating in spatial and spectral domains for the `FaceForensics++` data.

| Basis | Metrics | Classification on Spatial Features | | | | Classification on Frequency Features | | | | Proposed |
|---|---|---|---|---|---|---|---|---|---|---|
| | | Naskar et al. [19] | Agarwal et al. [1] | Das et al. [4] | He et al. [11] | Tan et al. [28] | Kohli et al. [15] | Jeong et al. [12] | Hasanaath et al. [10] | |
| In-Dataset Classification | Accuracy | 0.9701 | 0.9762 | 0.9905 | 0.9850 | 0.9350 | 0.9075 | 0.9471 | 0.9434 | 0.9493 |
| | Precision | 0.9719 | 0.9755 | 0.9884 | 0.9824 | 0.9280 | 0.9048 | 0.9447 | 0.9431 | 0.9487 |
| | Recall | 0.9702 | 0.9760 | 0.9891 | 0.9831 | 0.9311 | 0.9061 | 0.9453 | 0.9515 | 0.9502 |
| | F1-Score | 0.9706 | 0.9755 | 0.9885 | 0.9794 | 0.9295 | 0.9055 | 0.9440 | 0.9392 | 0.9495 |
| | Complexity | $\mathcal{O}\left(n \cdot d^2 \cdot T\right)$ | $\mathcal{O}\left(n \cdot d \cdot L^2 \cdot T\right)$ | $\mathcal{O}\left(n \cdot L \cdot h^2 + L \cdot a \cdot h^2\right)$ | $\mathcal{O}\left(n \cdot d \cdot k^2 \cdot T\right)$ | $\mathcal{O}\left(n \cdot d \cdot f \cdot T\right)$ | $\mathcal{O}\left(n \cdot d \cdot k^2 \cdot T\right)$ | $\mathcal{O}\left(n \cdot d \cdot f^2 \cdot k^2 \cdot T\right)$ | $\mathcal{O}\left(n \cdot d \cdot f \cdot L \cdot T\right)$ | $\mathcal{O}\left(n \cdot d \log d \cdot f\right)$ |
| Cross-Dataset Classification | Accuracy | 0.9401 | 0.9232 | 0.9615 | 0.9457 | 0.8816 | 0.8640 | 0.8745 | 0.8879 | 0.8883 |
| | Precision | 0.9354 | 0.9206 | 0.9579 | 0.9418 | 0.8782 | 0.8613 | 0.8720 | 0.8848 | 0.8876 |
| | Recall | 0.9387 | 0.9216 | 0.9594 | 0.9421 | 0.8758 | 0.8589 | 0.8689 | 0.8810 | 0.8875 |
| | F1-Score | 0.9365 | 0.9208 | 0.9585 | 0.9414 | 0.8741 | 0.8595 | 0.8669 | 0.8826 | 0.8876 |
| | Complexity | $\mathcal{O}\left(n^* \cdot d^2 \cdot T\right)$ | $\mathcal{O}\left(n^* \cdot d \cdot L^2 \cdot T\right)$ | $\mathcal{O}\left(n^* \cdot L \cdot h^2 + L \cdot a \cdot h^2\right)$ | $\mathcal{O}\left(n^* \cdot d \cdot k^2 \cdot T\right)$ | $\mathcal{O}\left(n^* \cdot d \cdot f \cdot T\right)$ | $\mathcal{O}\left(n^* \cdot d \cdot k^2 \cdot T\right)$ | $\mathcal{O}\left(n^* \cdot d \cdot f^2 \cdot k^2 \cdot T\right)$ | $\mathcal{O}\left(n^* \cdot d \cdot f \cdot L \cdot T\right)$ | $\mathcal{O}\left(n^* \cdot d \log d \cdot f\right)$ |

Table 2. Comparison of performance between different models operating in spatial and spectral domains for the `CelebDF (v2)` data.

| Basis | Metrics | Classification on Spatial Features | | | | Classification on Frequency Features | | | | Proposed |
|---|---|---|---|---|---|---|---|---|---|---|
| | | Naskar et al. [19] | Agarwal et al. [1] | Das et al. [4] | He et al. [11] | Tan et al. [28] | Kohli et al. [15] | Jeong et al. [12] | Hasanaath et al. [10] | |
| In-Dataset Classification | Accuracy | 0.9402 | 0.9644 | 0.9759 | 0.9704 | 0.9017 | 0.8808 | 0.9089 | 0.8950 | 0.9203 |
| | Precision | 0.9364 | 0.9598 | 0.9722 | 0.9683 | 0.8994 | 0.8780 | 0.9072 | 0.8916 | 0.9196 |
| | Recall | 0.9396 | 0.9620 | 0.9748 | 0.9694 | 0.9009 | 0.8788 | 0.9101 | 0.8937 | 0.9201 |
| | F1-Score | 0.9564 | 0.9609 | 0.9737 | 0.9685 | 0.8992 | 0.8785 | 0.9077 | 0.8925 | 0.9193 |
| | Complexity | $\mathcal{O}\left(n \cdot d^2 \cdot T\right)$ | $\mathcal{O}\left(n \cdot d \cdot L^2 \cdot T\right)$ | $\mathcal{O}\left(n \cdot L \cdot h^2 + L \cdot a \cdot h^2\right)$ | $\mathcal{O}\left(n \cdot d \cdot k^2 \cdot T\right)$ | $\mathcal{O}\left(n \cdot d \cdot f \cdot T\right)$ | $\mathcal{O}\left(n \cdot d \cdot k^2 \cdot T\right)$ | $\mathcal{O}\left(n \cdot d \cdot f^2 \cdot k^2 \cdot T\right)$ | $\mathcal{O}\left(n \cdot d \cdot f \cdot L \cdot T\right)$ | $\mathcal{O}\left(n \cdot d \log d \cdot f\right)$ |
| Cross-Dataset Classification | Accuracy | 0.9256 | 0.9138 | 0.9557 | 0.9371 | 0.8412 | 0.8174 | 0.8505 | 0.8330 | 0.8701 |
| | Precision | 0.9182 | 0.9100 | 0.9500 | 0.9314 | 0.8383 | 0.8144 | 0.8448 | 0.8294 | 0.8692 |
| | Recall | 0.9201 | 0.9124 | 0.9521 | 0.9338 | 0.8350 | 0.8126 | 0.8435 | 0.8257 | 0.8695 |
| | F1-Score | 0.9194 | 0.9115 | 0.9511 | 0.9335 | 0.8346 | 0.8118 | 0.8415 | 0.8243 | 0.8694 |
| | Complexity | $\mathcal{O}\left(n^* \cdot d^2 \cdot T\right)$ | $\mathcal{O}\left(n^* \cdot d \cdot L^2 \cdot T\right)$ | $\mathcal{O}\left(n^* \cdot L \cdot h^2 + L \cdot a \cdot h^2\right)$ | $\mathcal{O}\left(n^* \cdot d \cdot k^2 \cdot T\right)$ | $\mathcal{O}\left(n^* \cdot d \cdot f \cdot T\right)$ | $\mathcal{O}\left(n^* \cdot d \cdot k^2 \cdot T\right)$ | $\mathcal{O}\left(n^* \cdot d \cdot f^2 \cdot k^2 \cdot T\right)$ | $\mathcal{O}\left(n^* \cdot d \cdot f \cdot L \cdot T\right)$ | $\mathcal{O}\left(n^* \cdot d \log d \cdot f\right)$ |

Table 3. Ablation Study: Impact of DFT Filtering and Sub-band Energy Components on WAVEDIF's Accuracy

| Model Variant | DFT Filtering | $\mathcal{E}_{\mathrm{LL}}$ | $\mathcal{E}_{\mathrm{LH}}$ | $\mathcal{E}_{\mathrm{HL}}$ | $\mathcal{E}_{\mathrm{HH}}$ | Validation Accuracy | |
|---|---|---|---|---|---|---|---|
| | | | | | | FF++ [25] | CDF2 [17] |
| WAVEDIF | ✓ | ✓ | ✓ | ✓ | ✓ | 0.9493 | 0.9203 |
| w/o DFT Filtering | X | ✓ | ✓ | ✓ | ✓ | 0.8041 | 0.8049 |
| w/o $\mathcal{E}_{\mathrm{LL}}$ | ✓ | X | ✓ | ✓ | ✓ | 0.8333 | 0.8251 |
| w/o $\mathcal{E}_{\mathrm{LH}}$ | ✓ | ✓ | X | ✓ | ✓ | 0.8402 | 0.8390 |
| w/o $\mathcal{E}_{\mathrm{HL}}$ | ✓ | ✓ | ✓ | X | ✓ | 0.8411 | 0.8406 |
| w/o $\mathcal{E}_{\mathrm{HH}}$ | ✓ | ✓ | ✓ | ✓ | X | 0.8531 | 0.8459 |

each table, methods are compared based on achieved accuracy, precision, recall, and F1-score. Additionally, the complexities of the methods have been reported in terms of number of in-dataset samples ($n$), number of cross-dataset samples ($n^*$), feature dimension ($d$), number of epochs ($T$), number of layers ($L$), transformers' hidden dimension ($h$), convolutional networks' kernel size ($k$), transformers' attention heads ($a$), and frequency domain transformation complexity ($\mathcal{O}\left(f\right)$). The WAVEDIF pipeline (training) takes $n$ or $n^*$ data points, and each of them are converted to frequency domain in $\mathcal{O}\left(f\right)$ time. Wavelet decomposition of the filtered videos is done with $d \times \log d$ complexity (due to divide-and-conquer approach of Haar filters). Thus, overall, the proposed model's complexity is $\mathcal{O}\left(n \cdot d \log d \cdot f\right)$.

As observed through experiments (Table 1, and 2) – the proposed **WAVEDIF model outperforms state-of-the-art deepfake detection models (operating in frequency domain) for both in-dataset, and cross-dataset basis of test-ing by** $0.7433\%$**, and** $1.1748\%$ **respectively**. Table 3 gives an ablation analysis of each component in the WAVEDIF model with respect to both the datasets. Table 3 related to ablation study reveals that DFT filtering and all sub-band energy components ($\mathcal{E}_{\mathrm{LL}}$, $\mathcal{E}_{\mathrm{LH}}$, $\mathcal{E}_{\mathrm{HL}}$, $\mathcal{E}_{\mathrm{HH}}$) contribute to WAVEDIF's accuracy. Removing DFT filtering from the pipeline significantly lowers accuracy (by $\approx 15.31\%$), while excluding any sub-band component also reduces performance, with $\mathcal{E}_{\mathrm{LL}}$ having the largest impact ($\approx 12.23\%$).

## 5. Conclusion

This work introduced WAVEDIF, which uses wavelet sub-band energies for the detection of deepfake videos. Unlike traditional methods that use deep learning for feature extraction, WAVEDIF has a lightweight but efficient pipeline that uses DFT for high-frequency noise dropout, followed by DWT to extract sub-band energies as distinguishing features. WAVEDIF outperforms existing frequency (spectral) domain deepfake detection techniques, though like any other frequency-based approach, WAVEDIF falls short of its' accuracy compared to spatial domain methods. But, in contrast to the exponentially high count of model parameters needed to be trained for multi-modal methods, WAVEDIF have significantly lesser number of parameters, which also gets reflected in its' lightweight model complexity ($\mathcal{O}\left(n \cdot d \log d \cdot f\right)$). Future works can be conducted on the integration of temporal dynamics and adaptive thresholding to achieve generalizability across different deepfake generation methods.

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
