# OpenReview forum: "WaveDIF: Wavelet sub-band based Deepfake Identification in Frequency Domain"
_thecvf.com/CVPR/2025/Workshop/CVEU — CVPR 2025_

### Official Review · Reviewer_Ei5e · 2025-03-22
**Review of WAVEIDF: Wavelet sub-band based Deepfake Identification in Frequency Domain**

**Rating:** 5
**Confidence:** 4

**Review:**

Authors introduce a novel deepfake video detection algorithm operating strictly in the frequency domain.The algorithm extracts features using wavelet sub band energies.The process of first applying Discrete Fourier Transform (DFT) to filter out high-frequency noise from video frames and then decomposes the filtered representations into wavelet sub-bands and calculating their energy values is very interesting. These energy values are used to train a linear decision boundary for classifying videos as real or deepfake.

Deepfake is evolving and detection involves lof of challenges.Based on my experience, CNN based methodologies are dominant to solve this and are often effective for deepfake identification.However, they are computationally demanding.Authors proposed an alternative approach that compares very well with the existing methodologies.The research also demonstrate the effectiveness of WAVEDIF on deepfake datasets, showing comparable accuracy to state-of-the-art methods with the advantage of reduced computational cost.

## Strengths:

- Computation efficiency: A key strength of WAVEDIF is its computational efficiency.By using DWT and linear regression, the algorithm avoids the heavy computational demands of CNN-based methods.This is crucial for real-time deepfake detection.

- Interpretability: WAVEDIF offers a more interpretable approach compared to deep learning models.The feature extraction and classification processes are relatively transparent, allowing for a better understanding of how the algorithm makes decisions.

- Frequency Domain Focus: The paper effectively highlights the importance of the frequency domain in deepfake detection.It demonstrates that frequency artifacts can be discriminative features for identifying manipulated videos.

- Promising Results: The experimental results show that WAVEDIF achieves competitive accuracy on benchmark datasets, demonstrating its potential as a viable deepfake detection method.

- Clear Methodology: The paper provides a clear and well-defined explanation of the WAVEDIF algorithm, making it easy to understand and potentially reproduce.

## Weaknesses:

- Oversimplified Feature Representation: The use of wavelet sub-band energies, while efficient, might be an oversimplification.It is not clear and possible that this representation doesn't capture the full complexity of frequency artifacts present in deepfakes, potentially limiting the model's ability to detect more advanced manipulations.

- Lack of Robustness Evaluation: The evaluation provided primarily focuses on accuracy and doesn't thoroughly assess the robustness against adversarial attacks or its ability to generalize to unseen deepfake generation techniques.

- Limited Comparison with Hybrid Methods: While the paper compares WAVEDIF with spatial and frequency domain methods, it lacks a comprehensive comparison with hybrid methods that combine both domains. These hybrid methods often achieve state-of-the-art performance, and a more detailed comparison is needed to understand WAVEDIF's relative strengths and weaknesses.

- Thresholding Dependence: The classification relies on a threshold, which can be sensitive to data distribution and require careful tuning.The paper could explore alternative classification methods or discuss the limitations of threshold-based classification in more detail.

- Novelty: While the application of wavelet sub-band energies for deepfake detection is presented as novel, the individual components (DFT, DWT, regression) are well established techniques.The paper cuold more clearly articulate the specific novelty of their combination and application in this context.

## Areas for Improvement:

- Enhanced Feature Representation: Explore more sophisticated frequency domain features or consider a hybrid approach that combines wavelet sub-bands with other frequency domain representations to capture a richer set of artifacts.

- Robustness Testing: Conduct more rigorous testing to evaluate the model's robustness against adversarial attacks, novel deepfake generation methods, and variations in video quality.

- Practical Runtime Evaluation: Supplement the theoretical computational complexity analysis with empirical runtime comparisons against other methods, including CNN-based ones.

## Conclusion:

Overall, authors presented computationally effective approach called WAVEIDF for deepfake identification. The proposed approach of leveraging wavelet sub-band energies in the frequency domain sounds very interesting and worth discussion in workshop. While the paper demonstrates promising results, there are few areas where the methodology and evaluation could be strengthened. Addressing the identified weaknesses and incorporating the suggested improvements would enhance the paper's contribution and further validate the potential of WAVEDIF for practical deepfake detection applications. The approach presents an interesting discussion for workshop and aligns very well with scope and relevance of the workshop. Hence, i consider this as a strong accept and also recommend authors to address areas mentioned to strengthen their claims.

---

### Official Review · Reviewer_DLPV · 2025-03-25
**Review for WaveDiff,  a method for classify real videos from synethetically generated videos with wavelet transforms.**

**Rating:** 4
**Confidence:** 3

**Review:**

Summary:
In this paper the authors propose WaveDIF, a method for detecting AI generated videos. The proposed method uses wavelets and a simple regression to classify the videos. The relatively simplicity makes the method interpretable and lightweight.

Pros:
- The method is interesting. The results show that the method is able to achieve a good classification on the validation set.
- The simplicity of the method is appealing as it makes it interpretable and efficient to compute.

Cons:
- The method starts by computing the average DFT across every frame in the input video. This step is a little counter-intuitive to me.
- The method also applies a low pass filter before extracting the features. The motivation for this is also not that well explained.


Minor Comments:
- typo in caption from Fig 4 (“iff”)

---

### Official Review · Reviewer_5bHX · 2025-03-25
**Proposes a Lightweight Model, but Fails to Justify Its Advantages**

**Rating:** 2
**Confidence:** 4

**Review:**

This paper proposes a lightweight deepfake detection method. Its approach can be summarized as: (1) apply a Gaussian low-pass filter, (2) extract DFT and DWT features, and (3) train a linear classifier for binary prediction.

The novelty claims made by the authors (Lines 125–145) are not convincing.

Claim 1: While the proposed method differs from prior work, there already exist methods that operate in the frequency domain. The authors do not sufficiently justify the benefits of their approach. Even in the experimental results (Table 1), the best-performing methods are still based on spatial features.

Claim 2: Although deep learning-based methods can be computationally expensive, recent advances in specialized hardware and mixed-precision algorithms have made their execution much more efficient. The authors also fail to provide any experimental results to demonstrate the proposed method’s runtime advantage.

Claim 3: The authors claim that the proposed method is interpretable, but no actual interpretation is provided. Moreover, the large number of wavelet features extracted can be difficult to interpret in practice.

In addition, the experimental results in Tables 1 and 2 show that the proposed method performs comparably to other frequency-domain approaches, and is inferior to spatial-domain methods.

Based on these observations, I recommend a weak reject.

---

### Decision · Program_Chairs · 2025-03-25

**Decision:**

Accept

**Comment:**

The paper introduces WaveDIF, a computationally efficient and interpretable method for deepfake video detection using wavelet transforms in the frequency domain. Reviewers appreciate its simplicity, practical efficiency, and competitive results, though they highlight concerns about novelty and limited robustness evaluation. Overall, the strengths outweigh the weaknesses, making it suitable for acceptance at the workshop.